# Creating Understandable and Actionable COVID-19 Health Messaging for Refugee, Immigrant, and Migrant Communities

**DOI:** 10.3390/healthcare11081098

**Published:** 2023-04-12

**Authors:** Iris Feinberg, Mary Helen O’Connor, Saja Khader, Amy L. Nyman, Michael P. Eriksen

**Affiliations:** 1Adult Literacy Research Center, Georgia State University, Atlanta, GA 30302, USA; 2Prevention Research Center, Georgia State University, Atlanta, GA 30302, USA; 3School of Public Health, Georgia State University, Atlanta, GA 30302, USA

**Keywords:** health literacy, refugees, cultural and linguistic responsiveness, health communication, community-based participatory research

## Abstract

During the coronavirus pandemic, it was imperative that real-time, rapidly changing guidance on continuously evolving critical health information about COVID-19 be communicated. This case study highlights how understandable and actionable COVID-19 health information was systematically developed and disseminated to support highly vulnerable refugee, immigrant, and migrant (RIM) communities in Clarkston, Georgia. Our approach was grounded in community-based participatory research (CBPR) incorporating Cultural and Linguistically Appropriate Services (CLAS) standards, plain language and health literacy guidelines, and health communication science to improve the understandability and usability of COVID-19 micro-targeted messaging for RIM communities. We followed a centralized systematic approach to materials development and incorporated local needs and existing networks to ensure cultural and linguistic responsiveness as well as understandability for populations with limited literacy skills. Further, iterative development of materials with community members and agencies provided buy-in prior to dissemination. As part of a multi-pronged community-wide effort, effective materials and messaging provided support to community health workers and organizations working to improve vaccination rates among the RIM community. As a result, we saw vaccine rates in Clarkston outpace other similar areas of the county and state due to this community-wide effort.

## 1. Background

The coronavirus pandemic presented an urgent need to communicate real-time, rapidly changing guidance on constantly evolving critical health information about COVID-19, including mitigation strategies, testing, and eventually vaccine uptake [1]. The extensive amount of information, misinformation, and disinformation spread through traditional and social media exposed the health literacy gap between how messages are communicated and how they are understood and used to make informed health decisions [1,2]. The scarcity of trusted culturally and linguistically responsive messaging compounded the problem for refugee, immigrant, and migrant (RIM) populations at high risk of not being able to find, understand, or use health information during the COVID-19 pandemic [3,4]. Further, significant mistrust of the media, healthcare providers, government organizations, and public health agencies in many RIM communities was only exacerbated by the proliferation and quick dissemination of mixed messages on social media platforms [5,6,7,8]. RIM populations are disproportionately affected by ineffective risk communication practices that are not grounded in plain language, health literacy guidelines, and cultural and linguistic responsiveness [9].

Health literacy guidelines and cultural and linguistic responsiveness work in tandem to advance health equity and should be considered a universal standard, used with every communication, every time, for every person. As a two-sided construct, health literacy can be both an individual asset and an organizational responsibility; for RIM populations, limited English proficiency can significantly hinder health literacy skills, especially when health information is not shared in a culturally or linguistically responsive manner. For organizations that work with RIM populations, health literacy focuses on creating and sharing trustworthy, comprehensible and effective health information with vulnerable and high-risk communities [1]. Guidelines for cultural and linguistic responsiveness can be found in the United States Department of Health and Human Services’ Cultural and Linguistic Appropriate Services standards (CLAS), a blueprint to help providers of health information deliver understandable, respectful, and effective healthcare services that responds to diverse cultures, health beliefs, languages, and health literacy levels [10].

Clarkston, Georgia is a refugee resettlement community near Atlanta, Georgia where more than 60,000 refugees have initially settled since 1970 [11]. After the passage of the Refugee Act in 1980, the United States formalized the process of accepting and resettling refugees by establishing the Office of Refugee Resettlement (ORR) in the Department of Health and Human Services. ORR works with resettlement agencies (all NGOs) with branches and affiliate offices throughout the U.S. to provide support services to new arrivals. In the 1970s, Clarkston was identified by resettlement agencies as a community with a large supply of affordable housing and access to employment, education, and public transportation. Texas, California, New York, Minnesota, and Michigan also have large communities of resettled refugees in cities including Houston, San Diego, Sacramento, Buffalo, Minneapolis, and Grand Rapids.

Today, approximately half of Clarkston’s 17,000 residents are foreign-born and face significant and persistent substandard factors that undermine health including limited English proficiency, low health literacy, low literacy, high unemployment, low income, limited or no health insurance coverage, densely populated housing, and ineffective connection to federal and state-funded social support programs such as the Supplemental Nutrition Assistance Program (SNAP), Special Supplemental Nutrition Program for Women, Infants and Children (WIC), and Medicaid, a health insurance program for people with low income [3,10]. Refugees come to the U.S. with limited language skills, no financial resources, no guarantee of employment, and high levels of physical and mental stress. When they arrive, they are provided with a small, short-term package of support services funded by the U.S. government and provided through local resettlement agencies including housing assistance, English instruction, health insurance through Medicaid, and supplemental food programs such as SNAP and WIC. This support usually runs out within six to nine months, and individuals must maneuver through unfamiliar bureaucratic systems to re-apply for aid. Poor access to healthcare, limited English skills, and lack of transportation lead to almost 70% of Clarkston’s residents not knowing where to go or how to access these social support benefits [12].

Health communication messaging during the COVID-19 pandemic highlighted existing disparities and stressed how lack of language proficiency and cultural incongruence further marginalized vulnerable communities. Existing disparities such as the lack of digital access and poor digital literacy during a time of massive dissemination of internet-based health information further disregarded the needs of refugees and their families [3,10,13]. Stay-at-home orders and requests to wear masks also propagated mistrust, marginalization, fear, and stigma in a community accustomed to avoiding institutional authority because of flight from oppression and persecution in their home countries [14]. Further compounding the problem was the impact on refugees whose economic livelihoods were disproportionately impacted by COVID-19 restrictions, as manufacturing, service-based, and food processing industries are the primary employers for this population [15]. Many worked in frontline jobs as the sole provider for their family and could not work from home [15].

Healthy People 2030 stresses improvements in health communication so people can easily find, understand, and use health information to make informed health decisions for themselves and their families [16]. The ability to provide culturally and linguistically responsive health information is critical to the ongoing public health of the nation as the US population continues to become more diverse. Census data from 2020 indicate a rapid increase in diversity with Latino/Hispanic and Asian American populations growing by rates of 20% and 29%, respectively [17]. Clear, correct, concise, complete, concrete, and culturally responsive health information is one of the more modifiable factors contributing to health inequities [18]. To address this inequity and to respond to community needs to provide critical life-saving health information during the COVID-19 pandemic, we present a case study of Clarkston. Our aim was to use a systematic approach grounded by community-based participatory research (CBPR) principles to prioritize RIM perspectives on risk, health, and wellness, follow CLAS standards, and improve the understandability and usability in COVID-19 messaging by using plain language and health literacy guidelines.

## 2. Materials and Methods

### 2.1. Setting

Two non-profit, volunteer-led health clinics and one sliding scale clinic serve residents in the community along with the DeKalb County Board of Health with facilities in an adjacent city accessible by public transportation; the Federally Qualified Health Center serving the community is located several miles outside the city. As part of a CDC-funded Prevention Research Center (PRC) focused on social determinants of health in Clarkston, Georgia State University (GSU), researchers and community members formed an active Community Advisory Board (CAB) that includes local government and community members, clinicians, trusted community leaders, and representatives of community organizations [19]. When the COVID-19 pandemic began, it became immediately clear that a centralized approach to the pandemic in this multicultural and linguistically diverse community was needed. Health providers assembled a volunteer task force which met weekly, the Clarkston Community COVID-19 Task Force, whose members shared information, strategized, and collaborated on community-wide efforts to combat the pandemic. Task force activities were led by volunteer representatives from health clinics, local resettlement agencies, the PRC, local government, the local board of health, community residents, and academic partners.

### 2.2. Organizing Frameworks

Our approach to supporting this complex cultural, ethnic, and linguistic community was to integrate three frameworks to ensure inclusion of perspectives and needs of Clarkston’s RIM communities: following Community-Based Participatory Research (CBPR) principles, adhering to national standards for improvement of cultural and linguistic services (CLAS), and utilizing evidence-based health literacy guidelines as a universal precaution. CBPR is a translational approach that focuses on building community capacity and improving health outcomes by engaging the community as equal partners [20]. By supporting community members and organizations as the experts about their communities, we gained a deeper understanding and appreciation of how to design and disseminate health education. The 15 national standards for CLAS guide organizations to “provide effective, equitable, understandable, and respectful quality care and services that are responsive to diverse cultural health beliefs and practices, preferred languages, health literacy, and other communication needs” [10]; standards five to eight reference communication and language assistance for all populations served, including those who speak a language other than English. Health literacy universal precaution guidelines focus on ensuring that health messaging uses plain language, makes the healthcare system easier to navigate, and is used for every message, every time, and in every situation [2]. These guidelines simplify written materials so miscommunication is minimized, and support every person’s effort to improve their health [21]. Integration of these three frameworks allowed for precise and customized health communication for RIM communities that was clear, scientifically accurate, concise, concrete, complete, trustworthy, and culturally and linguistically responsive. Further, we followed four guiding principles of health communication science: identify the target behavior, deliver the message from trusted sources, develop strategies to expose people to the message, and take a comprehensive dissemination approach [22]. See Figure 1.

## 3. Creating and Disseminating Multicultural and Multilingual Materials

### 3.1. Process

Early in the pandemic, before vaccines were available and while not much more than basic mitigation strategies were known, the COVID-19 Community Task Force requested materials from the PRC to dispel common community-level myths and to provide instructions for prevention and testing behaviors in multiple languages. The materials development team consisted of faculty from the Adult Literacy Research Center (adult and health literacy subject matter expert) and the GSU Prevention Research Center (community engagement and refugee/migrant subject matter expert), and several student assistants (undergraduate and graduate) with lived experience of forced migration. 

Following guidelines from the CDC “Simply Put” guide for creating easy-to-understand materials [23], we gathered basic information from the COVID-19 Community Task Force that included topic, audience, and suggested dissemination modality. Draft messages were crafted to address the question or topic of concern by using plain language guidelines but always focusing on scientific accuracy. What we knew about mitigation strategies and protection against COVID-19 changed frequently, especially in the early days of the pandemic; we decided to start with CDC information, and then relied on the Task Force clinical providers to review and ensure scientific accuracy. To assure message clarity, we followed the Simply Put guidelines to (1) give the most important information first; (2) limit the number of messages; (3) describe what needed to be done; and (4) choose words carefully (avoid jargon and abbreviations, use few syllables, use general common words). Materials were designed using large, simple font (12–14 point, Times New Roman or Arial), dark ink on light background, adequate white space, and visuals that reflected the message and were culturally relevant. Each document was assessed for health literacy level using the Patient Education Materials Assessment Tool (PEMAT) [24] to ensure understandability and actionability in English.

Prior to translation, we reviewed the messages with our student assistants who were both members of the intended audience and English/foreign language speakers to ensure appropriate literacy level, cultural responsiveness, and understandability. One question we asked the students was “would your mother/father/aunt/uncle understand this message?” After meaningful discussions, content was revised as necessary, reviewed again, and then translated by either a community member or by a professional translator. The decision on translation was dependent on availability of typeset; for example, Swahili is written using English letters, however, Amharic is a character-based language and requires a keyboard or typewriter with appropriate alphabetic characters. Translators were paid for their time, and materials were reviewed once more by both students and other members of the intended audience to ensure correct translation and placement of words on the materials. Placement was a critical issue since some languages require all words of a phrase to be on one line whereas in English, each word of a phrase can be read on subsequent lines. 

Dissemination modality was the final consideration. Choices included printed flyers/brochures, social media posts, WhatsApp distribution lists, broader listservs, websites, and mass media options. We met with the Task Force to discuss distribution points for printed materials which included clinics, childcare centers, apartment complexes, and van-share to work locations. Social media posting included the PRC Instagram and Facebook channels; Task Force and CAB members offered to disseminate on their community and organization’s social media, websites, WhatsApp distribution lists, and appropriate listservs such as the state office of refugee services. 

### 3.2. Products

#### 3.2.1. Answers to Coronavirus Booklet

At the request of clinicians who were seeing patients with low literacy in their first language as well as limited English proficiency, the Task Force developed easily distributed paper-based booklets in English and seven other languages (Arabic, Amharic, Burmese, Nepali, Spanish, Swahili, Tigrinya) to address the most prevalent myths being heard in the community as well as by health providers. The myths ranged from “it’s just like getting a cold” to use of homemade remedies to prevent or treat COVID-19. In many cultures, “myth” can denote both positive and negative references [25]; thus, it was decided to avoid the use of the term altogether and instead frame the myth-dispelling sections as “This is what I heard” followed by the myth, then followed by the dispelling statement. For example: 

Myth: This is what I heard: Black people cannot get it.

Dispelling Statement: Anyone can get coronavirus. People of all colors have been very sick. Many have died. People of all colors have died. 

The second half of the booklet addressed other community concerns and mitigation strategies, specifically addressing some issues that were very important to the RIM community, including shame associated with not inviting friends over, i.e., social distancing, and stigma involved with testing. A critically important topic to address for these communities was what to do when leaving work, travelling home, and upon arriving home since many RIM community members are frontline essential workers, and were required to work in person every day. Hundreds of booklets were distributed in multiple languages at clinics, testing sites, childcare centers, and van-sharing to work locations. Printable booklets were also made available to the broader public through local and national listservs, and on several GSU research center websites. 

#### 3.2.2. Information Cards

We were asked to produce information cards to be distributed with personal protection equipment (PPE) at apartment complexes in Clarkston that were heavily populated with RIM families. The Task Force identified the limited economic ability of residents to obtain PPE and launched an effort to distribute masks, gloves, hand sanitizer, and health information door-to-door in high-density areas. We produced plain-language health information graphic-intensive cards included in the distribution of more than 20,000 PPE kits to community residents to instruct recipients on effective use of the kit including how to correctly wear a mask and where to get tested. Community and student volunteers canvassed apartment complexes on weekends to provide maximum coverage in the highest-risk areas of the city. 

#### 3.2.3. Protect Clarkston Lawn Signs

Lawn signs were created in partnership with Clarkston city government as part of the “Protect Clarkston” campaign; branding included using the city’s colors and ensuring appropriate partner logos. The driving concept behind the message was a message of collective responsibility as opposed to a more Western approach of individual responsibility, building on the more collectivist approaches in the cultures represented in Clarkston. The message was developed and vetted by representatives of the city government as well as through the existing process with community members. When vaccines were available, new lawn signs were made for a “Vaccinate Clarkston” campaign. Both Protect and Vaccinate Clarkston campaigns included 250 signs in six languages placed on highly trafficked routes in the city where most residents walk to public transportation, shopping centers, and places of worship. The signs remained in place throughout the pandemic.

#### 3.2.4. You Should Get a Vaccine

Once vaccines were available, we created 60 second “You should get a COVID-19 vaccine” animated videos for wide distribution. As with written materials, videos were vetted with language and culturally concordant community members to ensure cultural and linguistic responsiveness. The videos were translated into 20 languages and were disseminated widely through social media posts, WhatsApp distribution lists, broader listservs, and on websites.

#### 3.2.5. Vaccine Ambassador Flyers

As part of the CDC-funded PRC supplemental funding, a Vaccine Ambassador (VA) program was created within the Clarkston community, which targeted five ethnic/linguistic communities: speakers of Arabic, Swahili, Dari/ Pashto, Somali, and Burmese. A critical component to success was to ensure source reliability, that is, to ensure that community members viewed the VAs as trusted, reliable sources. VAs met with groups and individual community members, attended community health events, provided interpretation on-site at vaccination events, and canvassed the community to improve vaccine confidence. They frequently requested specific, targeted, simply-stated fact sheets to provide to their community members, most often in response to dispelling misinformation and disinformation, and addressing the emerging science around COVID-19 vaccines and boosters. Fact sheets were created for all communities with a variety of culturally responsive images (i.e., women in hijab, characters with a range of skin tones) and in different languages; however, other flyers were created specifically for needs expressed by that community. For example, RIM community members were often communicating messages they had received from home countries with political leaders overseas often referenced by the VAs as sources of misinformation. The now-deceased president of Tanzania denied the existence of the virus in his country, declaring that it had been defeated by prayer [26]. Many Clarkston community members are connected to family members still in-country; Tanzania also hosts many displaced individuals in camps as well as in the country. The deceased president’s message was rapidly shared via social media and mobile phone technology. Addressing misinformation from overseas is a continuous concern for this community. Twelve flyers unique to each community and each topic were created and disseminated widely. The VAs preferred printed flyers that they could hand to community members when meeting with them. Printed materials were also shared electronically through the social media channels and list-servs. See Appendix A for a sample flyer. 

#### 3.2.6. Bus Stop Campaign—Get Your Vaccine Today

In the continued effort to increase the Clarkston community’s confidence in and uptake of COVID-19 vaccines, we implemented a “Get Your Vaccine Today” poster campaign on Metropolitan Atlanta Rapid Transit Authority (MARTA) bus shelters in Clarkston featuring seven trusted community sources sharing a quote about why they got the COVID-19 vaccine. In a community where many residents rely on public transportation, trusted community leaders were invited to create a personalized message to encourage COVID-19 vaccination; messages were translated into multiple languages and placed in locations that were targeted to the language group that frequented that bus stop. The posters included a QR code with a link to the PRC’s website listing CORE Georgia’s (Community Organized Relief Effort) local weekly mobile vaccination sites schedules. CORE Georgia, in partnership with International Rescue Committee (IRC) and Dekalb County Board of Health, provides COVID-19 vaccines to everyone throughout the state of Georgia for free. The QR code was accessed 26 times in a four-week period; there were over 530,000 views of the bus stop posters during the same time. See Appendix A for a sample bus stop poster. 

## 4. Lessons Learned

The difficulty of communicating time-sensitive critical health information in a community where more than 60 languages are spoken is a persistent barrier to RIM health and wellness during a global pandemic. We learned lessons and used best practices applicable to communities from around the world where language, literacy, and cultural diversity among residents is a defining characteristic. Despite the urgent need for immediate health messaging, taking the time to develop and use a health literacy universal precautions approach to create accessible and trusted health information was perhaps the most critical lesson learned; this rigorous, systematic practice ensured appropriate materials for every cultural group in every situation every time, and was sustained throughout the pandemic. Community collaboration—whether with clinical providers or end-user community members—ensured scientific accuracy, ease of reading level, cultural sensitivity, and linguistic responsiveness, and created a trusted source for the community. Flexibility in a broad dissemination approach using many forms of messages and communication outlets delivered messages to people where they looked for and trusted health information. We continue these practices in current community-based projects addressing other health inequities and social determinants of health in Clarkston. 

Health information from government, public health, university-based, and health provider sources was rarely accessible to or understood by RIM community members with low literacy, limited English proficiency, and poor digital access/skills even if that information was translated into a non-English language. Materials from other refugee-serving sources often were translated; however, they rarely employed other health literacy guidelines to ensure message clarity, understandability, and actionability. Our community members and VAs often identified mistakes or requested revisions to widely shared materials from other public health and refugee-serving organizations because they were too complex or confusing to community residents. For example, few materials were designed using large, simple font with dark ink on light background with adequate white space, which are key factors in improving readability for people with limited literacy skills in any language. Other materials provided too many messages in one document, or did not describe what needed to be done, so actionability was not evident. Translation of English into any other language does not ensure understandability; we found words and phrases that did not translate directly. Input from community members was critical to ensuring cultural responsiveness and understanding.

The seriousness and urgency of COVID-19 combined with frequently changing traditional and social media messages created a sense of confusion for Clarkston community members and for those who worked with and supported them. The pandemic highlighted a plethora of accurate, complex, contradictory, and false information that was difficult to understand for non-native English-speaking people in Clarkston who had both low general literacy and health literacy skills as well as limited English proficiency [3]. Critical culturally and linguistically responsive public and individual health messaging that was accessible, understandable, and usable was urgently needed. We received high-priority requests from different sources, often multiple times a day, and sometimes with conflicting guidance. Using a culture-centered, participatory, and inclusive approach and combining expertise in evidence-based theory and practice from adult learning, general literacy, plain language, health literacy, community-based participatory research, and refugee studies brought an organized and meaningful approach to the translation of nuanced materials [27,28]. We followed a health literacy universal precautions model for all health communication, regardless of how simple or complex the message was [2]. 

Despite best efforts to provide easier-to-read and translated materials on both mitigation and vaccination, communication from public health agencies, government, health and healthcare providers, and academia was often not understood by members of the local RIM community. Professional formal language is often not familiar to the RIM populations who may use a more colloquial or conversational tone; thus, after creating messages using plain language and health literacy guidelines and obtaining approval by medical professionals, our materials were presented to community members from the corresponding language and cultural groups who were able to identify problematic words and phrases. For example, the word for COVID was identified by speakers of Swahili as “flu” and therefore was not understood as a different disease with its unique risk and transmission factors. Further, following guidelines for low-skilled readers and plain language, the most important information was given first; there were one or two messages presented, and each message included an action [23,29]. The source of the information was also crucial to uptake by community members. As opposed to government agencies, community members were more apt to follow guidelines and recommendations from local healthcare providers, members of their faith communities, and trusted community leaders [30]. 

While distanced and homebound, many populations depended on media, social media, and web-based communication to access information about the pandemic [31], but the digital divide was starkly exposed by COVID-19. Many RIM community members do not have smartphones, internet access, or adequate digital literacy and were stranded when trying to access services including education, health services, health information, and support services such as emergency benefits [3,12,32]. For the Clarkston RIM community, print materials were more effective in reaching community members than electronic media, as evidenced by the VA requests for the printing of thousands of flyers for direct distribution, the door-to-door PPE distribution, and the frequency of requests from community organizations for printed materials, distributed widely through local channels [32]. 

Members of marginalized and vulnerable groups such as RIM populations often have the least amount of access to resources during crises [33] and they often also have the greatest amount of knowledge and expertise about their communities [34]. Incorporating local needs and existing networks into materials development ensured targeted cultural and linguistic responsiveness; iterative development allowed for community-level agency and buy-in. Appreciating the community’s request for varied dissemination channels (print booklets, community flyers, handheld brochures, posters, WhatsApp, Instagram, Facebook, Twitter, and community email distribution lists) improved trust in the process and enhanced our ability to work quickly and efficiently in response to urgent community needs. Our goal was not to create translated readable materials; rather, the goal was to create understandable and culturally responsive materials. Understandability ensures that the person who is reading the material knows what to do the first time they read the message [35].

In summary, we were faced with a rapidly changing situation whereby providing timely, scientifically accurate, trustworthy, and actionable health information could change lives. By grounding all our work in the best practices of community engagement and applying health-literate evidence-based standards of practice, we were able to provide useful and meaningful health education materials throughout the COVID-19 pandemic. Despite this approach being expensive and time-consuming, applying these lessons learned during the pandemic is critical for any providers of health information: using CLAS, health literate approaches, and most importantly, a community-centered approach acknowledging the wide range of cultures, languages, and literacy levels to reach those community members who are the least able to access trusted and understandable information. It is critical to listen to the community before developing guidance. Without knowledge of what is being communicated and shared among various cultural groups, it would not be possible to address specific concerns in a way that is relevant. Having established relationships of trust is crucial to open and honest communication, allowing community residents to share their fears and concerns without the fear of judgment or derision. As part of a multi-pronged community-based effort, effective materials and messaging provided support to community health workers and organizations working to improve vaccination rates among the RIM community. In fact, vaccine rates in Clarkston outpaced other similar areas of the county and state due to this broad community-wide effort.

## Figures and Tables

**Figure 1 healthcare-11-01098-f001:**
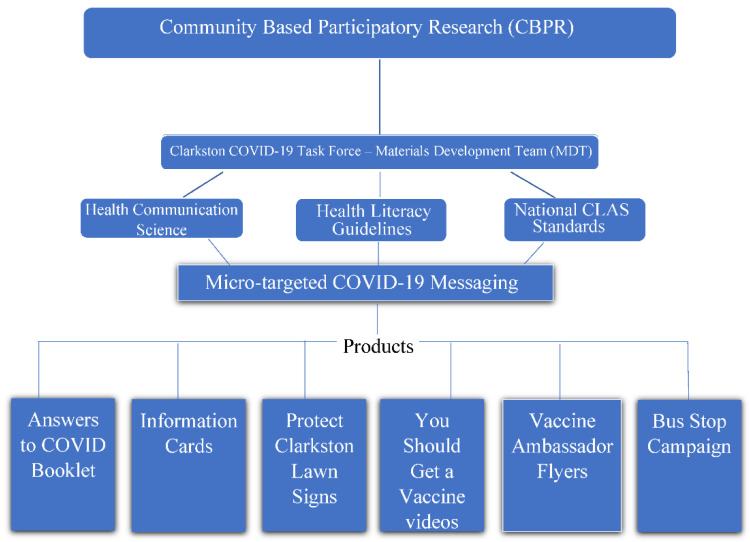
Process Diagram.

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
