# Peer review of "Creating Understandable and Actionable COVID-19 Health Messaging for Refugee, Immigrant, and Migrant Communities"

_healthcare, 2023, doi:10.3390/healthcare11081098_

Round 1

Reviewer 1 Report

Thank you so very much for the opportunity to review this case study. In a nutshell, I found this paper interesting, and it has the potential to offer useful insights for tailored public health messaging not only for emergencies like pandemics but also for regular interventions. That said, I recommend minor edits to the manuscripts and expanding a few sections before publishing.

1. Provide the full form of RIM and include implications of your research/case study and key takeaways in the abstract.

2. Line 35: Aren't RIMs considered "vulnerable"? Or, within RIM, are there certain sections vulnerable to which this sentence refers to? I suggest removing the "vulnerable" word if the first is the case or expanding what authors have to say if the latter is true.

3. Line 47: It is unclear by the word "national" which country the authors are referring to. Until line 51, readers do not convey the case study setting. Even after mentioning the place, the authors did not provide the country setting. For international readers, it will be easier if the country is specifically mentioned without asking them to make a guess.

4. Line 51-72: International readers might be curious about the history of RIM settlements in the USA, where they are settled throughout the country. Some background is needed here.

5. Line 57: provide long forms of SNAP and WIC. For the international audience, please describe SNAP, WIC, and Medicaid and why they are important for RIM populations.

8. The Theoretical Frameworks section needs to elaborate “health literacy universal precautions model” that authors have alluded to in the Lessons Learned section.

9. Lines 300,308,309: Citation numbers should be superscripted.

10. Lessons Learned Section: RIM communities have been settled throughout the USA, and several organizations are working with them. In this section, the authors should connect with those communities and organizations by discussing (dis)similarities in approaches during the pandemic and how and if (in)effective strategies were shared across the regions/states to tackle almost similar challenges.

11. A summary of the overall approach (what, how, and when) must be provided before the last paragraph of the Lessons Learned section.

Author Response

Reviewer 1

Thank you so very much for the opportunity to review this case study. In a nutshell, I found this paper interesting, and it has the potential to offer useful insights for tailored public health messaging not only for emergencies like pandemics but also for regular interventions. That said, I recommend minor edits to the manuscripts and expanding a few sections before publishing.

  1. Provide the full form of RIM and include implications of your research/case study and key takeaways in the abstract.

Response:  This has been expanded.

  1. Line 35: Aren't RIMs considered "vulnerable"? Or, within RIM, are there certain sections vulnerable to which this sentence refers to? I suggest removing the "vulnerable" word if the first is the case or expanding what authors have to say if the latter is true.

Response: we have removed the word vulnerable in this sentence.

  1. Line 47: It is unclear by the word "national" which country the authors are referring to. Until line 51, readers do not convey the case study setting. Even after mentioning the place, the authors did not provide the country setting. For international readers, it will be easier if the country is specifically mentioned without asking them to make a guess.

Response: this has been corrected

  1. Line 51-72: International readers might be curious about the history of RIM settlements in the USA, where they are settled throughout the country. Some background is needed here.

            Response:  We have added some more information about this topic. 

  1. Line 57: provide long forms of SNAP and WIC. For the international audience, please describe SNAP, WIC, and Medicaid and why they are important for RIM populations.

            Response: added background here

  1. The Theoretical Frameworks section needs to elaborate “health literacy universal precautions model” that authors have alluded to in the Lessons Learned section.

            Response: we have expanded the definition in this section.

  1. Lines 300,308,309: Citation numbers should be superscripted.

            Response: This has been corrected.

  1. Lessons Learned Section: RIM communities have been settled throughout the USA, and several organizations are working with them. In this section, the authors should connect with those communities and organizations by discussing (dis)similarities in approaches during the pandemic and how and if (in)effective strategies were shared across the regions/states to tackle almost similar challenges.

            Response: This is a very interesting comment. During these critical times, we found that everyone was “peddling as fast as they could” to support RIM individuals.  There was little time to reflect or share.  When opportunities arose to do so, unfortunately, other national RIM-serving organizations took a tunnel visioned approach about their work and there was little interest in learning lessons about others’ work. For example, the NCRIM at University of Minnesota collected a repository of materials but in our discussions with them, none of the materials were evaluated for health literacy, actionability, understandability, or usability by the end user community which, for our community, made the repository unusable.

  1. A summary of the overall approach (what, how, and when) must be provided before the last paragraph of the Lessons Learned section.

Response: This has been done.

Reviewer 2 Report

- In the abstract, more specific methods should be described. In the abstract, spell out RIM.

- Introduction: the first two sentences require proper citations.

-

Although I appreciate the researchers' efforts to develop and disseminate COVID-19-related health messages to immigrant and refugee groups, I am skeptical that this manuscript in its current form is academically valuable. It is difficult to know, for example, whether the development of the health message (intervention) was based on the communication of health-related theories. As figures, the authors should provide examples of their intervention. The authors stated that they attempted to incorporate culturally and linguistically compatible materials, but it is difficult to know without examples. Furthermore, it is unclear whether their health message (intervention) was validated. It's also unclear how these messages were received by immigrants and refugees.

Author Response

Reviewer 2

  1. In the abstract, more specific methods should be described. In the abstract, spell out RIM.

            Response: We have extended the abstract.

  1. Introduction: the first two sentences require proper citations.

            Response: We added a citation to the first sentence; the second sentence is already cited.

3, Although I appreciate the researchers' efforts to develop and disseminate COVID-19-related health messages to immigrant and refugee groups, I am skeptical that this manuscript in its current form is academically valuable.

            Response: Thank you for your comments.  We address each of them individually.

  1. It is difficult to know, for example, whether the development of the health message (intervention) was based on the communication of health-related theories.

Response: Apologies, but we are not clear as to what this question is trying to ask. If you are asking if we used health behavior and promotion theories to develop the materials, (such as the Theory of Planned Behavior), we did not.  We used health literacy guidelines and health communication techniques and theories.

  1. As figures, the authors should provide examples of their intervention. The authors stated that they attempted to incorporate culturally and linguistically compatible materials, but it is difficult to know without examples.

Response: Excellent idea, we have attached two figures of interventions that were created.

  1. Furthermore, it is unclear whether their health message (intervention) was validated. It's also unclear how these messages were received by immigrants and refugees.

Response: We do not intend for this case study to be a research study with testing and validation of materials. We did not have time to test the health messages to ensure validity either because the need was changing or the science itself was changing so quickly. For scientific content, we only used science provided by the CDC; for materials creation we followed evidence-based practices in adult literacy, health literacy, and CLAS standards; for CBPR, we followed theoretical guidelines. 

Acceptance by the community can only be supported as part of a multi-pronged community-wide effort, providing support to community health workers and organizations working to improve vaccination rates among the RIM community. We saw vaccine rates in Clarkston outpace other similar areas of the county and state due to this community-wide effort.

Reviewer 3 Report

This case report is considered to be a very valuable report that shares the health messaging strategy for the RIM communities, where it was difficult to properly respond to COVID-19 due to weak access to health information. Nonetheless, I would like to make a few comments about this manuscript.

- In the abstract, it is recommended to briefly and additionally describe the production and dissemination of multicultural and multilingual materials corresponding to the outcome of this study. In addition, a description of the meaning and suggestion on the results of this study is required.

- It seems necessary to present an accurate term for RIM in the abstract.

- If possible, please supplement the description of the actual effect of the achievements of this report in the conclusion (4. Lessons Learned). For example, in this report, it would be nice to present a part on whether the multicultural and multilingual materials developed and disseminated were correlated with the incidence of COVID-19 or the awareness of RIM in the region.

Author Response

Reviewer 3

This case report is considered to be a very valuable report that shares the health messaging strategy for the RIM communities, where it was difficult to properly respond to COVID-19 due to weak access to health information. Nonetheless, I would like to make a few comments about this manuscript.

  1. In the abstract, it is recommended to briefly and additionally describe the production and dissemination of multicultural and multilingual materials corresponding to the outcome of this study. In addition, a description of the meaning and suggestion on the results of this study is required. It seems necessary to present an accurate term for RIM in the abstract.

            Response: We have expanded the abstract.

  1. If possible, please supplement the description of the actual effect of the achievements of this report in the conclusion (4. Lessons Learned). For example, in this report, it would be nice to present a part on whether the multicultural and multilingual materials developed and disseminated were correlated with the incidence of COVID-19 or the awareness of RIM in the region.

Response: We did not study an actual effect and cannot state correlation between materials and vaccine rates.  However, we do know that as part of a multi-pronged community-wide effort, effective materials and messaging provided support to community health workers and organizations working to improve vaccination rates among the RIM community. In fact, vaccine rates in Clarkston outpaced other similar areas of the county and state due to this broad community-wide effort.

Round 2

Reviewer 2 Report

Although the authors revised the manuscript well, taking into account the reviewers' comments, I believe that more presentations of findings with scientific evidence are required.

Author Response

Although the authors revised the manuscript well, taking into account the reviewers' comments, I believe that more presentations of findings with scientific evidence are required.

RESPONSE:  This was not a rigorous scientific study.  It is a case presentation.  I am afraid we cannot provide any more scientific findings because we do not have them.  We provide a case study as per one of the options called for in the special call.